# Equity-Informative Economic Evaluations of Vaccines: A Systematic Literature Review

**DOI:** 10.3390/vaccines11030622

**Published:** 2023-03-09

**Authors:** Chanthawat Patikorn, Jeong-Yeon Cho, Philipp Lambach, Raymond Hutubessy, Nathorn Chaiyakunapruk

**Affiliations:** 1Department of Pharmacotherapy, College of Pharmacy, University of Utah, Salt Lake City, UT 84112, USA; 2Department of Social and Administrative Pharmacy, Faculty of Pharmaceutical Sciences, Chulalongkorn University, Bangkok 10540, Thailand; 3School of Pharmacy, Sungkyunkwan University, Suwon 16419, Republic of Korea; 4Department of Immunization, Vaccines and Biologicals (IVB), World Health Organization, 1211 Geneva, Switzerland; 5IDEAS Center, Veterans Affairs Salt Lake City Healthcare System, Salt Lake City, UT 84112, USA

**Keywords:** equity, inequality, disparity, economic evaluation, cost-effectiveness analysis, vaccine, immunization

## Abstract

The Immunization Agenda 2030 prioritizes the populations without access to vaccines. Health equity has been increasingly incorporated into economic evaluations of vaccines to foster equitable access. Robust and standardized methods are needed to evaluate the health equity impact of vaccination programs to ensure monitoring and effective addressing of inequities. However, methods currently in place vary and potentially affect the application of findings to inform policy decision-making. We performed a systematic review by searching PubMed, Embase, Econlit, and the CEA Registry up to 15 December 2022 to identify equity-informative economic evaluations of vaccines. Twenty-one studies were included that performed health equity impact analysis to estimate the distributional impact of vaccines, such as deaths averted and financial risk protection, across equity-relevant subgroups. These studies showed that the introduction of vaccines or improved vaccination coverage resulted in fewer deaths and higher financial risk benefits in subpopulations with higher disease burdens and lower vaccination coverage—particularly poorer income groups and those living in rural areas. In conclusion, methods to incorporate equity have been evolving progressively. Vaccination programs can enhance equity if their design and implementation address existing inequities in order to provide equitable vaccination coverage and achieve health equity.

## 1. Introduction

The number of children not receiving a single dose of routine vaccine (defined as the first dose of diphtheria, tetanus, and pertussis (DTP1) non-receipt), also referred as “zero-dose children”, increased by 5 million in 2021 compared with 2019, going from 13 to 18 million. More than 60% of these children live in extremely poor conditions facing a lack of access to reproductive health services, water, and sanitation [1]. Health equity has been increasingly incorporated into economic evaluations of vaccines to foster equitable access. The Immunization Agenda 2030 prioritizes populations that are not being reached through current immunization efforts—particularly the most marginalized communities, those living in fragile and conflict-affected settings, mobile populations, and those moving across borders [2]. Robust and standardized methods are needed to evaluate the health equity impact of vaccination programs to ensure monitoring and effective addressing of inequities.

The Immunization Agenda 2030, through its Strategic Priority 3, addresses equity by defining key areas of focus and objectives to reach the goal of protecting everyone with full immunization, regardless of location, age, socioeconomic status, or gender-related barriers [2]. The World Health Organization’s “Guide for Standardization of Economic Evaluations of Immunization Programmes” also recommends that the health equity impact be included if it is considered an important factor for decision-making [3]. These recommendations emphasized the need to explore and summarize how health equity was incorporated and evaluated in the existing literature on economic evaluations of vaccines.

Health technology assessment has been employed in many countries to inform healthcare decision-making [4]. This is especially relevant to countries aiming to provide accessible, affordable, equitable, and high-quality healthcare services to their populations while ensuring the sustainability of health systems in place. Equity-informative assessments can provide data on the health equity impact of health technologies and public health policies and the inherent tradeoff between total coverage and equitable coverage. Based on these data, decision-makers can better balance the efficient use of limited budgets and foster equitable access to healthcare. Health equity impact analysis has been increasingly incorporated into the economic evaluations of health technologies and public health policies, including vaccines [5,6,7,8,9,10,11,12,13]. Health equity impact analysis is conducted to estimate the distribution of impact of alternative policy options, broken down by one or more variables of concern to policymakers from an equity perspective [14]. Nevertheless, varying methods to evaluate the health equity impact can affect the application of findings to inform policy decision-making.

Several systematic reviews summarize equity-informative economic evaluations in terms of methodological aspects and the application of the methods in general [5,15,16,17,18,19]. However, no systematic review comprehensively describes how health equity is incorporated into the economic evaluations of vaccines. In addition, economic evaluations of vaccines differ from other health technologies given the unique characteristics of vaccines, such as program deployment costs, vaccination coverage, and herd protection [3]. Therefore, we conducted a systematic literature review to identify economic evaluations of the health equity impact of vaccines and immunization programs, focusing on the methods and applications.

## 2. Materials and Methods

The protocol of this review was registered with PROSPERO (CRD42022382729). We reported this review following the 2020 Preferred Reporting Items for Systematic Reviews and Meta-analyses (PRISMA) [20]. The PRISMA checklist table of this review is provided in Appendix A.

### 2.1. Search Strategy and Eligibility Criteria

We searched for equity-informative economic evaluations of vaccines in electronic databases, including PubMed, Embase, Econlit, and Cost-Effectiveness Analysis (CEA) Registry by Tufts Medical Center from database inception to 15 December 2022. The search terms used included a combination of vaccine, economic evaluation, and equity terms, which were modified to match the search techniques of each database. No language restriction was applied. We also screened reference lists of eligible articles to identify further potentially eligible articles. A full search strategy is presented in Appendix A.

We included articles that met the following eligibility criteria: full-text articles of economic evaluations estimating costs, outcomes, and health equity impact of vaccines across equity-relevant subgroups in any context. After duplicates were removed, identified articles were independently screened and selected by two reviewers (C.P. and J.-Y.C.) using the eligibility criteria. Article selection was performed using EndNote 20.3. Disagreements were resolved with consensus by discussing with the third reviewer (N.C.)

### 2.2. Data Extraction

Two reviewers (C.P. and J.-Y.C.) independently extracted data from the selected studies using the data extraction form developed and pilot-tested based on five randomly chosen articles to finalize the form. Discrepancies in data extraction were resolved with consensus by discussing with the third reviewer (N.C.).

The following data were extracted from the selected articles: first author, year of publication, country, vaccine, equity-relevant subgroups, existing inequities, intervention(s) and comparator(s), perspective, measurement of health and non-health benefits, model type, the inclusion of herd protection, and study findings, including cost-effectiveness and health inequity impact of vaccines.

### 2.3. Quality Assessment

Two reviewers (C.P. and J.-Y.C.) independently performed reporting quality assessment using the Consolidated Health Economic Reporting Standard (CHEERS) 2022 statement [21]. Any disagreements during the reporting quality assessment were resolved by consensus upon discussion with the third reviewer (N.C.).

### 2.4. Data Synthesis

Following data extraction, we summarized how health equity was incorporated and evaluated in the selected economic evaluations of vaccines, including methodological characteristics, characteristics of vaccines and immunization programs, existing inequities in the health systems, characteristics of equity-relevant subpopulations, and study findings. Equity-relevant subpopulations were categorized following the PROGRESS-Plus framework, including (1) place of residence, (2) race/ethnicity/culture/language, (3) occupation, (4) gender/sex, (4) religion, (5) education, (6) socioeconomic status, (7) social capital, (8) personal characteristics associated with discrimination (e.g., age, disability), (9) features of relationships (e.g., smoking parents, excluded from school), and (10) time-dependent relationships (e.g., leaving the hospital, respite care, other instances where a person may be temporarily at a disadvantage) [22].

## 3. Results

### 3.1. Study Selection

A database search identified 613 records, of which 19 articles met the eligibility criteria [6,7,8,9,10,11,12,13,23,24,25,26,27,28,29,30,31,32,33]. Citation searching of the eligible articles further identified two articles [34,35]. Thus, twenty-one articles were included in this review. These articles were published in 2011 and later. The study selection flow is presented in Figure 1. Excluded studies based on full-text assessment are shown with reasons for exclusion in Appendix A.

### 3.2. Study Characteristics

Characteristics of the included studies are summarized in Table 1 and Table 2, as well as Appendix A. Studies were performed in many regions of the world, with most studies conducted in Sub-Saharan African countries (*n* = 8, 38%) [6,7,8,11,12,23,24,33], and six of them were performed in Ethiopia [6,7,8,11,12,33]. A large proportion of studies focused on vaccination programs in low- and middle-income countries (LMICs) (*n* = 17, 81%) [6,7,8,9,10,11,12,13,23,24,27,28,29,30,33,34,35]. These studies included a total of 11 antigens, of which rotavirus was commonly evaluated (*n* = 11, 52%) [10,11,13,23,27,28,29,30,33,34,35], followed by human papillomavirus (HPV) (*n* = 5, 24%) [9,12,25,26,35] and *Streptococcus pneumoniae* (*n* = 4, 19%) [8,31,32,35]. Rotavirus vaccine was the most commonly studied in LMICs (11 out of 17 studies, 65%) [10,11,13,23,27,28,29,30,33,34,35], while HPV vaccines [25,26] and pneumococcal vaccination [31,32] (two out of four studies, 50% each) were the most commonly studied vaccines in HICs. The breakdown of antigen by income economy is shown in Appendix A.

### 3.3. How Equity Has Been Incorporated into Equity-Informative Economic Evaluations of Vaccines

#### 3.3.1. Overall Methods

All studies were cost-effectiveness analyses that performed health equity impact analyses to estimate the distributional impact of vaccines across equity-relevant subpopulations of interest (Table 2, with details in Appendix A). Eleven studies performed only health equity impact analysis as part of cost-effectiveness analyses to estimate the distributional impact and subpopulation incremental cost-effectiveness ratios (ICERs) of vaccines [23,24,25,26,27,28,29,30,31,32,34]. Nine studies are Extended Cost-Effectiveness Analyses that performed health equity impact analysis of vaccines with an estimation of the distributional financial risk protection [6,7,8,9,10,11,12,13,35]. One study is a Distributional Cost-Effectiveness Analysis that performed a health equity impact analysis of vaccines, incorporating equity-weighting and opportunity costs as the money was displaced to be spent on vaccines instead of other health services [33]. All studies used static models, of which herd protection of vaccines was considered in a base-case analysis in one study [25] and in a scenario analysis in four studies [10,13,32,34].

#### 3.3.2. Existing Inequities across Equity-Relevant Subpopulations

These analyses were designed to simulate the distributional impact of vaccines within the existing health inequities across the equity-relevant subpopulation in the context of interest, where there were differences between more or less socially disadvantaged subpopulations. Existing inequities in these studies were inequities in disease mortality (*n* = 17, 81%) [7,8,9,11,13,23,25,26,27,28,29,30,31,32,33,34,35], vaccination coverage (*n* = 12, 57%) [7,8,11,12,23,25,27,28,29,30,33,35], disease incidence/prevalence (*n* = 11, 52%) [6,7,12,24,25,26,31,32,33,34,35], and financial risk (*n* = 9, 43%) [6,7,8,9,10,11,12,13,35].

Equity-relevant subpopulations of interest were socioeconomic status (*n* = 11, 52%) [6,7,8,9,10,11,12,13,27,33,35], race/ethnicity (*n* = 3, 14%) [26,31,32], and place of residence (regions, states, or rural/urban areas) (*n* = 2, 10%) [24,34]. The other five studies assessed the combination of characteristics of equity-relevant subpopulations (socioeconomic status, race/ethnicity, place of residence, and gender) [23,25,28,29,30], such as estimating the distributional effect of rotavirus vaccine across rural/urban areas, regions, gender, and income quintiles in India [28].

Socioeconomic status was categorized as income quintiles [6,7,8,9,10,11,12,13,23,27,28,29,30,33,35] or tertiles [25], ranging from the poorest to the richest. Income quintiles were defined using an asset index [23,28,29,30], gross domestic product per capita, Gini coefficient [8,13,35], and the National Demographic Health Survey [10,12]. However, some studies did not report how socioeconomic status was defined [6,7,9,11,25,27,33]. Regions were categorized following the National Demographic Health Survey [23,28,30]. There was no clear description of how rural and urban areas were defined [24,28,29].

#### 3.3.3. Vaccination Programs Evaluated

Intervention(s) and comparator(s) assessed in the economic evaluations were mostly between the introduction of a vaccination program vs. no vaccination (*n* = 12, 52%) [6,8,9,10,11,13,24,26,28,31,34,35]. The remaining studies were modeled to evaluate the distributional impact of improving vaccination coverage of the vaccination programs across equity-relevant subpopulations. These included the introduction of a vaccine into a routine vaccination program vs. the introduction of a vaccine into a routine vaccination program with improving vaccination coverage vs. no vaccination (*n* = 3, 14%) [23,29,30], improving vaccination coverage vs. status quo of the currently implemented vaccination program (*n* = 4, 19%) [7,12,32,33] and improving vaccination coverage vs. status quo vs. no vaccination (*n* = 2, 10%) [25,27].

Strategies to improve equitable vaccination coverage described in four studies can be categorized into two broad approaches. Firstly, strategies specifically designed to improve vaccination coverage in the more socially disadvantaged groups, including investing additional resources into rotavirus vaccine delivery in rural areas [33], providing financial incentives for those who received measles vaccine as part of routine immunization with the aim to increase vaccination coverage by 10% in the bottom two income quintiles [7], and revising the eligibility criteria of receiving pneumococcal vaccination to increase the number of eligible vaccine recipients, especially in the Black population in the US [32]. Secondly, strategies designed to achieve equal vaccination coverage across equity-relevant subpopulations, including providing supplemental doses of measles vaccine in addition to the doses prescribed in the standard vaccination schedule (i.e., supplementary immunization activities (SIAs) or mass campaigns) with the aim to achieve 90% vaccination coverage in all income quintiles [7] and providing HPV vaccine as a school-only program or implementing a new mandatory law requiring active opting-out of HPV vaccination with equal coverage across ethnicity and income tertiles [25].

Potential benefits of achieving equitable vaccination coverage were also estimated in four studies, of which two studies estimated the impact of incremental reductions in vaccine under-coverage from current to full coverage [23,29]. The other studies investigated the impact of a scenario when all equity-relevant subpopulations had the same vaccination coverage as the highest coverage subpopulation [27,30]. However, these studies did not describe how to achieve the said equitable vaccination coverage.

#### 3.3.4. Health and Non-Health Benefits of Vaccination Programs

Outcomes captured in these studies were chosen according to the health and non-health benefits of a particular vaccine to demonstrate the distributional impact of vaccination programs across equity-relevant subpopulations. The health benefits of vaccines included the prevention of deaths [6,7,8,9,11,13,23,27,28,29,30,31,32,33,34,35], cases [12,24,26,31,32], hospitalizations and outpatient/clinic visits [10,34], disability-adjusted life years (DALYs) [23,27,28,29,30,34], the gain in years of life saved [26], quality-adjusted life years (QALYs) [25,31,32], and health-adjusted life years (HALYs) [33].

Non-health benefits of vaccines, captured specifically in extended cost-effectiveness analyses, were quantified as financial risk protection in terms of household out-of-pocket (OOP) expenditures averted [6,7,8,9,10,11,12,13], catastrophic health expenditures (CHE) averted [6,10,12], the money-metric value of insurance [8,13], and impoverishment averted [10,35]. The definitions and components of financial risk protection differed across studies. For example, CHE was defined differently across three studies. CHE was defined as a proportion of disease-related expenditure exceeding a specific threshold of household income or expenditures, including 10% of monthly household income [10], 40% of total household consumption expenditures [12], and 10% of total household consumption expenditures or 40% of non-food total household consumption [6]. Impoverishment was defined as household income falling below the World Bank poverty line [35] or country-specific poverty line due to medical expenditures [10]. The money-metric value of insurance or risk premium was defined as the difference between the expected value of the individual’s income and the income the individual is willing to have in order to have an outcome that is certain [8,13].

### 3.4. Summary of Study Findings on Cost-Effectiveness and Health Equity Impact

The cost-effectiveness and health equity impact findings of vaccines are summarized in Appendix A. Subpopulation ICERs were estimated in ten studies that found similar findings of better cost-effectiveness results (lower ICERs) in equity-relevant subpopulations with higher disease burdens, especially the poorer-income groups and rural areas [23,24,25,27,28,29,30,31,32,34]. This demonstrated that introducing vaccines or improving vaccination coverage, compared to no vaccination, was more cost-effective in the more socially disadvantaged groups.

We found similar findings of more deaths averted and higher financial risk protection benefits in subpopulations with higher disease burdens, such as poorer income groups and those living in rural areas, across 21 studies [6,7,8,9,10,11,12,13,23,24,25,26,27,28,29,30,31,32,33,34,35]. However, higher household OOP expenditures were averted more in the wealthier income groups due to the aversion to private healthcare utilization [8,9,11].

Studies estimating the distributional impact of improving [7,25,31,33] or achieving [23,28,29,30] equitable vaccination coverage found that more deaths were averted in the more socially disadvantaged groups with higher disease burdens and lower vaccination coverage. Furthermore, one distributional cost-effectiveness analysis demonstrated that the pro-poor vaccination strategy of the rotavirus vaccine compared to the currently implemented program was a “lose-win” strategy as it showed a negative impact on total health despite a positive impact on health equity, which required a trade-off between efficiency and equity [33]. Interestingly, one study found that introducing rotavirus vaccine in the context of existing inequities in vaccination coverage across regions and socioeconomic subpopulations resulted in introducing disparities in the mortality reduction [23].

### 3.5. Reporting Quality

Reporting quality of the included studies, assessed using the CHEERS 2022 statement [21], is presented in Appendix A. Overall, most topics were adequately reported in the included studies. However, the health economic analysis plan and engagement with patients and others affected by the study were not reported in any study.

## 4. Discussion

Economic evaluations are typically performed to estimate the average incremental costs and effectiveness of interventions of interest. Equity-informative economic evaluations further provide a spectrum of impact across equity-relevant subpopulations to inform policy prioritization. This systematic review identified 21 equity-informative economic evaluations of vaccination programs to date, with progressively evolving methods to incorporate equity. The health equity impact of vaccines has been incorporated into economic evaluations by estimating the distributional health and non-health benefits of vaccination programs across equity-relevant subpopulations to better understand where and to whom more efforts and support should be provided. Extended cost-effectiveness analyses of vaccines were generally performed in LMICs to reflect the importance of financial risk protection, which is one of the goals of the health system for achieving universal health coverage [6,7,8,9,10,11,12,36,37]. Distributional cost-effectiveness analyses of vaccines were performed to estimate the distribution of health opportunity costs [33,38]. Since a vaccination program generally involves a large cohort of the population, distributional cost-effectiveness analyses could inform the trade-offs between improving total population health and reducing health inequities.

Existing inequities related to vaccines were shown in the included studies, where disease burden and financial risk were generally higher in more socially disadvantaged groups. There was usually lower vaccination coverage in poorer income quintiles, along with higher disease incidence and mortality compared to richer income quintiles. Successfully implemented equitable vaccination programs could help decrease diseases, deaths, and costs to health systems and households, as we found that immunization programs informed by equity-informative economic evaluations of vaccines generally resulted in more deaths averted and higher financial risk protection benefits in socially disadvantaged subpopulations compared to regular immunization programs [7,25,31,33]. Thus, equity-informed vaccination programs could enhance access to life-saving immunization for disadvantaged populations and ultimately help achieve health equity, if specifically designed to address existing inequities in health systems.

Forceful national and global decision-making on how best to adapt and optimize the implementation of immunization programs to reach all vaccination target groups needs to be underpinned by robust and standardized equity-informative economic evaluations. To ensure the ubiquitous application of such evaluations, global guidance is needed to incorporate health equity into economic evaluations and to ensure standardization in conducting, reporting, and interpreting the analyses. In this review, we highlight a few methodological considerations on how to shape future equity-informative economic evaluations of vaccines. Firstly, the health equity impact of improving vaccination coverage should be conducted to provide information on the potential benefits of moving towards achieving equitable vaccination coverage across equity-relevant subpopulations. Many studies were conducted to estimate the impact of vaccines introduced to contexts with existing inequities in vaccination coverage without consideration of the potential benefits of equitable vaccination coverage. Hence, models should be developed considering improving vaccination coverage as a gradual change rather than an instantaneous change to fully capture the marginal benefits of improving vaccination coverage. Different levels of target vaccination coverage should also be explored to develop evidence-informed optimal implementation strategies, as attempts to improve coverage early on (e.g., from 10% to 20%) are expected to have higher marginal benefits compared to boosting coverage in contexts with existing higher vaccination coverage (e.g., from 75% to 85%).

Secondly, we emphasize the importance of incorporating and reporting all relevant aspects of equity, as improving equity in one aspect could potentially lead to inequities in other aspects. For example, a pro-poor vaccination program that improved equitable vaccination coverage can introduce disparities in mortality reduction given the existing inequities in the mortality risk at baseline. Thus, policy decision-makers will be well-informed about both the positive and negative impacts of the vaccination programs.

Thirdly, a dynamic model should be developed to fully capture the distributional impact of most vaccination programs on the force of infection in susceptible individuals and indirect transmission-dependent effects [3]. Nevertheless, it is challenging to model herd protection between equity-relevant subpopulations—for example, modeling how higher vaccination coverage among the richer income groups will translate to herd protection for the unvaccinated in the poorer income groups.

Lastly, as highlighted by the CHEERS 2022 statement [21], stakeholder engagement is important to ensure that the studies align with needs of local stakeholders and policy decision-makers. De facto, none of the included studies reported the inclusion of stakeholder engagement. Thus, advocacy is needed to ensure that stakeholder engagement is included and transparently reported in future equity-informative economic evaluations of vaccines. Likewise, the inclusion of stakeholder engagement in economic evaluations, especially local stakeholders, is highly encouraged to gain a better understanding of their needs, opinions, and perceptions of how health equity and inequities are defined, measured, monitored, interpreted, and achieved. This is particularly important as we found that assessment and measurement of health equity impact were affected when equity was not clearly defined.

We accentuated a few limitations of our review that are worth mentioning. First, no specific guidelines or checklists are available to directly evaluate the equity-relevant methodological quality of equity-informative economic evaluations. Thus, quality assessment of the included studies could be carried out only in terms of reporting quality. Furthermore, the implications and applications of this review should be carefully interpreted since its findings and conclusions were based on a limited number of equity-informative economic evaluations of vaccines published since 2011. Analytical techniques of incorporating health equity in economic evaluations are continuously evolving, and we expect more studies to be published in the future.

## 5. Conclusions

The health-equity impact of vaccination programs has been increasingly estimated in economic evaluations across equity-relevant subpopulations to portray and/or address existing health inequities in health systems. Vaccines can enhance equity if the design and implementation of vaccination programs incorporate the effort and strategies to address existing health inequities to provide equitable vaccination coverage and achieve health equity. Guidelines on incorporating health equity into economic evaluations need to be developed to ensure standardization in conducting, reporting, and interpreting the analyses.

## Figures and Tables

**Figure 1 vaccines-11-00622-f001:**
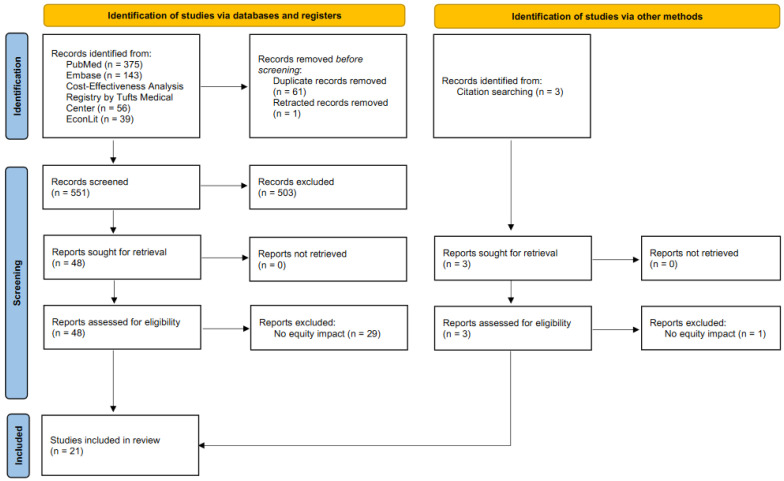
Study selection flow diagram.

**Table 1 vaccines-11-00622-t001:** Summary of included studies.

	Health Equity Impact Analysis (*n* = 11)	Health Equity Impact Analysis with Financial Risk Protection (*n* = 9)	Health Equity Impact Analysis with Equity Weighting (*n* = 1)	Total (*n* = 21)
**Region**				
Sub-Saharan Africa	2	5	1	8 (38%)
East Asia and Pacific	2	2	-	4 (19%)
North America	3	-	-	3 (14%)
South Asia	2	-	-	2 (10%)
Latin America and Caribbean	1	-	-	1 (5%)
Multiple countries	1	2	-	3 (14%)
**Income economy**				
High-income	4	-	-	4 (19%)
Low- and Middle-income	7	9	1	17 (81%)
**Antigen ***				
Rotavirus	6	4	1	11 (52%)
Human papilloma virus	2	3	-	5 (24%)
Streptococcus pneumoniae	2	2	-	4 (19%)
Malaria ^†^	1	1	-	2 (10%)
Measles	-	2	-	2 (10%)
Hepatitis B	-	1	-	1 (5%)
Hemophilus influenzae type b	-	1	-	1 (5%)
Yellow fever	-	1	-	1 (5%)
Rubella	-	1	-	1 (5%)
Neisseria meningitidis serogroup A	-	1	-	1 (5%)
Japanese encephalitis	-	1	-	1 (5%)

Note: * Number of studies may not add up, as some included multiple vaccines. ^†^ Malaria vaccine (RTS,S/AS01).

**Table 2 vaccines-11-00622-t002:** Methodological characteristics of included studies.

	Health Equity Impact Analysis (*n* = 11)	Health Equity Impact Analysis with Financial Risk Protection (*n* = 9)	Health Equity Impact Analysis with Equity Weighting (*n* = 1)	Total (*n* = 21)
**Equity-relevant subgroups**				
Socioeconomic status	1	9	1	11 (52%)
Race/Ethnicity	3	-	-	3 (14%)
Place of residence	2	-	-	2 (10%)
Combination of characteristics	5	-	-	5 (24%)
**Existing inequities ***				
Mortality	10	6	1	17 (81%)
Vaccination coverage	6	5	1	12 (57%)
Disease incidence/prevalence	6	4	1	11 (52%)
Financial risk	-	9	-	9 (43%)
**Intervention(s) vs. Comparator(s)**				
Introduction vs. No vaccination	5	7	-	12 (57%)
Introduction vs. Introduction with improving vaccination coverage vs. No vaccination	3	-	-	3 (14%)
Improving vaccination coverage vs. Status quo	1	2	1	4 (19%)
Improving vaccination coverage vs. Status quo vs. No vaccination	2	-	-	2 (10%)
**Perspective of analysis ^†^**				
Societal (Health system and household)	1	8	-	9 (43%)
Health system	10	-	1	11 (52%)
Household	-	1	-	1 (5%)
**Costs ***				
Direct medical costs	11	9	1	21 (100%)
Direct non-medical costs	1	8	-	9 (43%)
Indirect costs	1	3	-	4 (19%)
**Measurement of health benefits ***				
*Outcomes averted*				
Deaths averted	8	7	1	16 (76%)
DALYs averted	6	-	-	6 (29%)
Cases averted	4	1	-	5 (24%)
Hospitalizations and outpatient/clinic visits averted	1	1	-	2 (10%)
*Outcomes gained*				
QALYs gained	3	-	-	3 (14%)
HALYs gained	-	-	1	1 (5%)
Years of life saved	1	-	-	1 (5%)
**Measurement of financial risk protection ***				
Household OOP expenditures averted	-	8 ^‡^	-	8 (38%)
Catastrophic health expenditures averted	-	3	-	3 (14%)
Money-metric value of insurance (risk premium)	-	2	-	2 (10%)
Impoverishments averted	-	2	-	2 (10%)
**Model type**				
Dynamic	-	-	-	0 (0%)
Static	11	9	1	21 (100%)
**Herd protection**				
Included in base-case analysis	1	-	-	1 (5%)
Included in scenario analysis	2	2 ^§^	-	4 (19%)
Not included	8	7	1	16 (76%)

Abbreviations: DALY—disability-adjusted life year; HALY—health-adjusted life year; OOP—out-of-pocket; QALY—quality-adjusted life year. Note: * Number of studies may not add up, as some used multiple approaches. ^†^ Perspective was categorized based on authors’ statements in the articles or reviewers’ judgment based on methodologies of the studies. ^‡^ Two studies also estimated financial risk protection as household OOP expenditures averted as a percentage of household income. ^§^ Distributional effect of herd protection was estimated across subpopulations.

## Data Availability

The data presented in this study are available within this article and its Appendix A.

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
