# Peer review of "Equity-Informative Economic Evaluations of Vaccines: A Systematic Literature Review"

_vaccines, 2023, doi:10.3390/vaccines11030622_

Round 1
Reviewer 1 Report
I have reviewed this informative article. The quality of the abstract needs improvement.
This study describes that health equity has increasingly been incorporated and evaluated in the economic evaluations of vaccines. The Immunization Agenda 2030 prioritizes the populations that are not currently being reached. To ensure monitoring and effective addressing of inequities, robust, standardized methods are needed to evaluate health equity impact. However, methods currently in place vary, potentially affecting the application of findings to inform policy decision-making. To identify equity-informative economic evaluations of vaccines, we performed a systematic review by searching PubMed, Embase, Econlit, and CEA Registry up to December 15, 2022.
Please revise the article and remove minor English grammar errors. I suggest the authors take English editing services from some agencies to improve the quality of this study. I am suggesting some studies. Please read these studies and improve your article.
Introduction section
I suggest that authors to read the suggested studies and add the latest citations to the introduction, literature and method sections to enhance the quality of the study.
Moradi, F., Ziapour, A., Najafi, S., Rezaeian, S., Faraji, O., . . . Soroush, A. (2021). Comparing the Associated Factors on Lifestyle Between Type 2 Diabetic Patients and Healthy People: A Case-Control Study. Int Q Community Health Educ, 272684X211022158. doi:10.1177/0272684X211022158
Geng, J., Ul Haq, S., Ye, H., Shahbaz, P., Abbas, A., & Cai, Y. (2022). Survival in Pandemic Times: Managing Energy Efficiency, Food Diversity, and Sustainable Practices of Nutrient Intake amid COVID-19 Crisis. Frontiers in Environmental Science, 13, 945774. doi:10.3389/fenvs.2022.945774
Literature section:
Add literature section. You cannot delete this section. Read the suggested literature studies to enhance your work's quality. Add a few lines about studies on how education and social media can educate people.
Farzadfar, F., Naghavi, M., Sepanlou, S. G., Saeedi Moghaddam, S., Dangel, W. J., Davis Weaver, N., . . . Larijani, B. (2022). Health system performance in Iran: a systematic analysis for the Global Burden of Disease Study 2019. The Lancet, 399(10335), 1625-1645. doi:10.1016/S0140-6736(21)02751-3
Su, Z., Cheshmehzangi, A., Bentley, B. L., McDonnell, D., Segalo, S., Ahmad, J., . . . da Veiga, C. P. (2022). Technology-based interventions for health challenges older women face amid COVID-19: a systematic review protocol. Syst Rev, 11(1), 271. doi:10.1186/s13643-022-02150-9
Materials and Methods
This section is very weak. Please follow the suggested studies and improve your paper. The authors need to improve this section. I am recommending some good studies. Read the methods of these studies, and improve your paper. Suggested useful articles citations:
Hafeez, A., Dangel, W. J., Ostroff, S. M., Kiani, A. G., Glenn, S. D., . . . Mokdad, A. H. (2023). The state of health in Pakistan and its provinces and territories, 1990–2019: a systematic analysis for the Global Burden of Disease Study 2019. The Lancet Global Health, 11(2), e229-e243. doi:https://doi.org/10.1016/S2214-109X(22)00497-1
Schmidt, C. A., Cromwell, E. A., Hill, E., Donkers, K. M., Schipp, M. F., Johnson, K. B., . . . Hay, S. I. (2022). The prevalence of onchocerciasis in Africa and Yemen, 2000-2018: a geospatial analysis. BMC Med, 20(1), 293. doi:10.1186/s12916-022-02486-y
Result
Read the results of these studies, and improve your paper according to these studies in this section. Suggested useful articles citations
Micah, A. E., Bhangdia, K., Cogswell, I. E., Lasher, D., Lidral-Porter, B., Maddison, E. R., . . . Dieleman, J. L. (2023). Global investments in pandemic preparedness and COVID-19: development assistance and domestic spending on health between 1990 and 2026. The Lancet Global Health. doi:https://doi.org/10.1016/S2214-109X(23)00007-4
Discussion section:
The separate heading of the discussion section should be around one page. Improve the study and make it strong. See the recommended studies and improve your sections.
Conclusion
Highpoint creativity and scientific contribution of this study to the body of literature. The English level needs corrections to meet scientific merit for publication. I accept and endorse this manuscript for publication after minor corrections, as suggested.
Reviewer 2 Report
The authors performed an outstanding study that adds valuable information to the topic.
Figure 1 should be narrowed or displayed horizontally.
The supplementary material is also well prepared and presented.
Author Response
Thank you for your suggestion. We have adjusted figure 1 to be readable.
Reviewer 3 Report
This review aims to identify economic evaluations of the health equity impact of vaccines and immunization programs, focusing on the methods and applications.
The review is interesting and I have only minor suggestions.
The strength of this review is the PROSPERO registration. Moreover, the authors provided a full description of the criteria used to perform the literature review. Nevertheless, the PRISMA flow diagram is not readable. Please, check it.
The discussion section is a bit redundant with the results sections. The authors should reduce the reference to their results, improving the comparison with international papers.
minor points:
- please, check the format of the 3.3.3 headline;
- please, check the character type used in the text, as well as the use of the short form.
Reviewer 4 Report
The article is very interesting. It is well planned and developed. No methodological errors found. It is only necessary to review a few small issues.
1) The fonts used for the introduction, and other parts of the paper are smaller that the used in the rest of the paper, I don’t see any reason for that. Please unify the font size.
2) In Line 142 the authors introduce de acronym LIMC without explaining it. There are many other like DTP Review the whole text It is best practice to spell out the full term before using the acronym or abbreviation, and then use the shortened form thereafter.
3) The English of the paper should be reviewed to make it more easy to read. Some examples are included below but the authors need to review it in deep.
· In the abstract there are two confuse sentences “To foster equitable access, health equity” and “To ensure monitoring and effective addressing of inequities, robust, standard…” Please Rewrite both sentences to avoid a dangling modifier.
· In the abstract the sentence “Vaccination programs can be equity-enhancing if their design and implementation address the existing inequities with the aim of providing equitable vaccination coverage and eventually achieving health equity” may be unclear or hard to follow. Consider rephrasing to “Vaccination programs can be equity-enhancing if their design and implementation address the existing inequities to provide equitable vaccination coverage and achieve health equity.”
· There are also two sentences with Dangling modifiers “To foster equitable access, health equity has been increasingly incorporated and evaluated (…) “ and “[2]. To ensure monitoring and effective addressing of inequities, robust, standardized methods are needed to evaluate (…)”Please rewrite both sentences because there are Dangling modifiers that make the test difficult to understand.
· Change “decision-makers are enabled to better balance the efficient use of limited budgets a” for “decision-makers can better balance the efficient use of limited budgets
· The sentence “Impoverishment was defined as where household income would fall below the World Bank poverty line or country-specific poverty line due to medical expenditures(...)” is hard to follow change into” Impoverishment was defined as household income falling below the World Bank poverty line or country-specific poverty line due to medical expenditures.”
· Please replace “Health equity impact of vaccines has been incorporated into economic evaluations by estimating the distributional health and non-health benefits of vaccination programs across equity-relevant subpopulations to gain a better understanding of where and to whom more efforts and support should be provided.].” By the following sentence
· “The health equity impact of vaccines has been incorporated into economic evaluations by estimating the distributional health and non-health benefits of vaccination programs across equity-relevant subpopulations to understand better where and to whom more efforts and support should be provided
· Please change “Extended cost-effectiveness analyses of vaccines were generally performed in LMICs to reflect the importance of financial risk protection which is one of the goals of health systems pursuing to achieve universal health coverage [6-12, 36, 37” for “Extended cost-effectiveness analyses of vaccines were generally performed in LMICs to reflect the importance of financial risk protection, which is one of the health systems' goals to achieve universal health coverage [6-12, 36, 37]”
· Replace “To ensure ubiquitous application of such evaluations, global guidance is needed to incorporate health equity into economic evaluations and to ensure standardization on how to conduct, report, and interpret the analyses” by “To ensure ubiquitous application of such evaluations, global guidance is needed to incorporate health equity into economic evaluations and to ensure standardization on conducting, reporting, and interpreting the analyses.”
Reviewer 5 Report
Estimated Editors in Vaccines,
in this very interesting and well-written systematic review, Patikorn et al address the topic of Equity-informative economic evaluations of vaccines. Their study is based on 21 primary researches, whose content is summarized across the paper. Unfortunately, no quantitative assessment was made because of the high heterogeneity of the studies, and this issue severely affects the potential significance of this research.
However, the present reviewer finds this study of more than decent quality, across the whole of its content. Therefore, I'm endorsing the acceptance of this study as it was submitted to Vaccines.
Author Response
Thank you for your time reviewing our manuscript.Round 2
Reviewer 1 Report
Accept in present form